# Nonlinear Vibration Characteristics and Bifurcations of a Rotor System Subjected to Brush Seal Forces

Yingyong Zou [1,2], Mukai Wang [3], Duhui Lu [3,*], Yongde Zhang [1], Zili Xu [4] and Yeyin Xu [4]

1   School of Mechanical and Power Engineering, Harbin University of Science and Technology, Harbin 150080, China; zouyy@ccu.edu.cn (Y.Z.)
2   School of Mechanical and Vehicle Engineering, Changchun University, Changchun 130022, China
3   Key Laboratory of Special Equipment Safety and Energy-Saving for State Market Regulation, China Special Equipment Inspection & Research Institute, Beijing 100029, China
4   School of Aerospace Engineering, Xi'an Jiaotong University, Xi'an 710049, China; xuyeyin@xjtu.edu.cn (Y.X.)
*   Correspondence: luduhui@126.com; Tel.: +86-010-5906-8231; Fax: +86-010-5906-8231

**Abstract:** In the paper, nonlinear vibration characteristics of a rotor system are investigated. Such a nonlinear rotor system is subjected to brush seal forces, which are obtained by integrating the bristle force along the entire ring. The nonlinear brush seal rotor system is constructed by merging a flexible rotor with nonlinear seal forces. The research is aimed at studying the nonlinear vibration characteristics and bifurcations of the motions under a variety of eccentricity circumstances. Different kinds of bifurcations are successfully obtained by mathematical discretization and mapping manipulation. Such a discrete mapping method successfully predicts the stable and unstable motions accurately. The period-doubling bifurcations and saddle node bifurcations of the rotor system are obtained. The sole unstable solutions are obtained, which are special, and a normal numerical integration method cannot solve this problem, which provides advantages in rotor design and motion control. According to the results, nonlinear resonances are found between the stable and unstable motions. The greater the eccentricity of the rotor, the greater the number of bifurcation points that occur during the rotor's nonlinear motions, as well as the larger the ranges of speeds where the motions are unstable. Saddle node bifurcations generate unstable nonlinear motions and non-smooth motions, which may bring damage to the mechanical rotors. The period-doubling bifurcations produce the route from period-1 to period-2 motions in the nonlinear rotor system. The research provides a new perspective to study the bifurcations and stability of the nonlinear rotor systems.

**Keywords:** bifurcations; nonlinear motions; semi-analytical method; nonlinear rotor





## 1. Introduction

Rotating machinery with brush seals is an effective and efficient equipment where the brush seal can reduce leakage and improve the vibration performance. Brush seals do not require the shaft and the seal rings to be centrally symmetrical, so the brush seals not only increase the system's thermal efficiency but also its steadiness and stability. It is one of the most economic and practical seals, and has various applications [1–4]. To continue to improve the design and efficiency of the brush seal, it is important to figure out the impacts of nonlinear seal forces on the rotor dynamics. One determines the seal force by a combination of experiments and theoretical analysis. Sun et al. [5–7] constructed a brush seal rotor experimental bench to examine the bristle deformation and motions of the brush seal. They found that the motion primarily occurred in the area where the bristle density was increased when the pressure was increased, and the free end of the bristle usually oscillated. A 3D heat transfer model of the brush seal was developed in order to conduct the investigation of the thermodynamic properties of the brush seal rotor system. It was difficult to produce an exact model of the seal force that could be used in the test since the brush bristles had complex working phases [8]. Researchers suggested

approximate theoretical models to determine the seal forces [9–12]. Due to nonlinearities, the influence of the brush seals on the rotor system has to be studied. Chai et al. [13] performed a study of the brush seal leakage rate and established a theoretical model of the brush seal rotor system considering flow–thermal–structure coupling. Wei et al. [14,15] constructed a multi-stage rotor system with radial offset of the disk. Nonlinear seal forces were derived analytically. The Runge–Kutta method was employed to study the nonlinear response of the brush seal rotor system. They optimized the parameters, and this resulted in a more stable rotor system. Amer et al. [16] adopted the perturbation method of multiple scales to deal with a nonlinear rotating system and obtained nonlinear solutions with good accuracy. Ha et al. [17] applied numerical simulation to study the effects of the installation position of the porous brush bristle model on the dynamics of the brush seal rotor system. They found that the direct stiffness, cross-coupling stiffness, and damping of the brush seal installed upstream were lower than those downstream. Zhang et al. [18] built a nonlinear brush seal rotor system based on a nonlinear Darcian porous media. The rotor dynamics with various pressure ratios and inlet spinning rates were evaluated.

In this paper, an analytical brush seal force model is established and the nonlinear dynamic characteristics of brush seal rotor system are studied by a semi-analytical method for accurate stability and bifurcations. This paper is structured as follows: the first section is the introduction. The second section establishes the dynamic model of the brush seal rotor system, which includes the nonlinear brush seal force of a single bristle. The resultant seal force model of the full ring and the brush seal rotor system are built then. The nonlinear motions and bifurcations are achieved in the third section by a semi-analytical technique. Nonlinear motion illustrations of the brush seal rotor system are presented. Finally, the stabilities are evaluated and discussed.

## 2. The Nonlinear Rotor System

A brush seal rotor system is established from a flexible rotor with nonlinear brush seal forces. The discrete mapping method [19,20] is used to transform the governing equations into a discretized nonlinear system. Consider the brush seal rotor system as

$$\mathbf{M}\ddot{\mathbf{X}} + \mathbf{C}\ddot{\mathbf{X}} + \mathbf{K}\dot{\mathbf{X}} = \tilde{\mathbf{F}}_s + \mathbf{F}_\omega + \mathbf{G} \tag{1}$$

where $\mathbf{M} = \mathrm{diag}(m, m)$ is the mass matrix of the nonlinear rotor system; $\mathbf{C} = \mathrm{diag}(c, c)$ is the damping coefficient matrix; $\mathbf{K} = \mathrm{diag}(k, k)$ is the stiffness matrix; $\tilde{\mathbf{F}}_s = [\overline{F}_x, \overline{F}_y]^{\mathrm{T}}$ is the nonlinear brush seal force; $\mathbf{F}_\omega = me\omega^2[\cos \omega t, \sin \omega t]^{\mathrm{T}}$ is the centrifugal force of the nonlinear rotor system; $\mathbf{G} = [0, mg]^{\mathrm{T}}$ is the weight vector. $\mathbf{X} = [X, Y]^{\mathrm{T}}$ represents the displacements in $x$- and $y$-directions. $e$ is the mass eccentricity and $\omega$ is the angular speed.

On the rotor system, nonlinear forces are introduced by the brush seal, which is mounted on the frame around the rotor journal. Elasticity and beam theory are utilized to determine the force that a single brush bristle exerts on the rotor. The contact force between the bristle and the rotor can be derived in accordance with the following assumptions: (1) it is assumed that the contact between the bristle and the rotor is always in a point-to-point contact and the contact point is at the end point of the bristle; (2) the bristle is always in elastic deformation. The contact diagram that depicts the interaction of a single brush bristle and the rotor is illustrated in Figure 1, and the parameters are in Table 1.

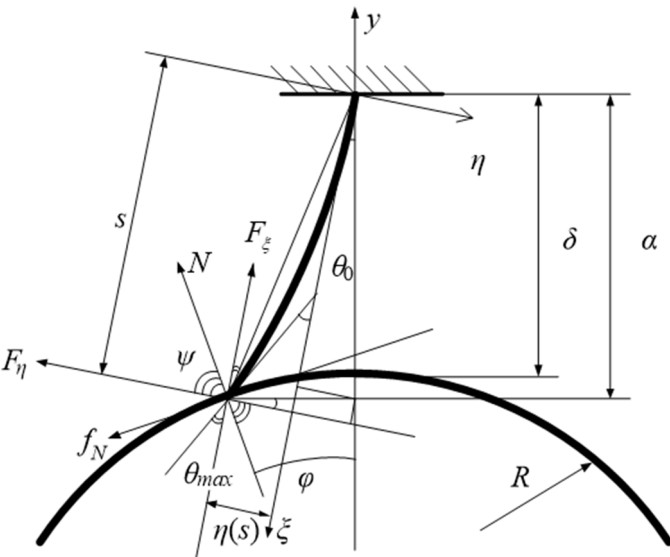

**Figure 1.** Contact diagram of the bristle and the rotor.

**Table 1.** Brush seal parameter nomenclature.

| Name | Symbols |
|---|---|
| Brush bristle deformation coordinate system | $(\eta, \xi)$ |
| Brush seal coordinate system | $(x, y)$ |
| Bristle length after deformation | $s$ |
| Actual radius clearance | $\delta$ |
| Bristle contact deflection angle | $\varphi$ |
| The $\eta$-directional force of the bristle | $F_\eta$ |
| The $\xi$-directional force of the bristle | $F_\xi$ |
| Maximum turning angle of the bristle | $\theta_{max}$ |
| Sliding friction | $f_N$ |
| The projection of $s$ on the radius direction | $\alpha$ |
| Shaft radius | $R$ |
| Bristle pre-rotation angle | $\theta_0$ |
| Maximum deflection | $\eta_{(s)}$ |
| Normal force on shaft journal | $N$ |
| Normal force of a single brush bristle on the shaft journal | $N_1$ |
| Friction coefficient of the rotor | $\mu$ |
| Angle between normal force and sealing force | $\psi$ |
| Eccentricity | $e$ |
| Declination angle | $\gamma$ |
| Minimum gap | $\delta_{max}$ |
| The angle between the circumferential direction and the negative direction of the y-axis | $\rho$ |
| Brush bristle diameter | $R_0$ |
| Moment of inertia of the bristle cross section | $I$ |

From Figure 1, consider the friction force between the bristle and the rotor as $f_N = \mu N$. Then, the contact forces $F_\eta$ and $F_\xi$ in the $\eta$- and $\xi$-directions are obtained as:

$$F_\eta = N \cos \psi + f_N \sin \psi = \sqrt{1 + \mu^2} N \sin(\theta_0 + \varphi + \arctan\mu),$$
$$F_\xi = N \sin \psi - f_N \cos \psi = \sqrt{1 + \mu^2} N \cos(\theta_0 + \varphi + \arctan\mu)$$

(2)

In Equation (2), $F_\xi$ is much smaller than $F_\eta$, so $F_\xi$ is neglected.

Consider the bristle in the bending diagram in Figure 2. Based on elasticity and beam theory, we have:

$$\theta(\xi) = -F_\eta \frac{\xi^2}{2EI} + \frac{F_\eta s}{EI}\xi,$$
$$w(\xi) = -F_\eta \frac{\xi^3}{6EI} + \frac{F_\eta s}{EI}\frac{\xi^2}{2}. \tag{3}$$

where $I = \pi R_0^4/4$. Considering the small deformation of the bristle, the relationship between the force on the bristle and the deformation can be obtained as.

$$L = \int_0^s \frac{1}{\cos\theta}\mathrm{d}\xi = \int_0^s 1 + \frac{\theta^2(\xi)}{2}\mathrm{d}\xi = s + \frac{F_\eta^2}{2E^2I^2}\int_0^s\left(-\frac{\xi^2}{2} + s\xi\right)^2\mathrm{d}\xi = s + \frac{F_\eta^2}{15E^2I^2}s^5 \tag{4}$$

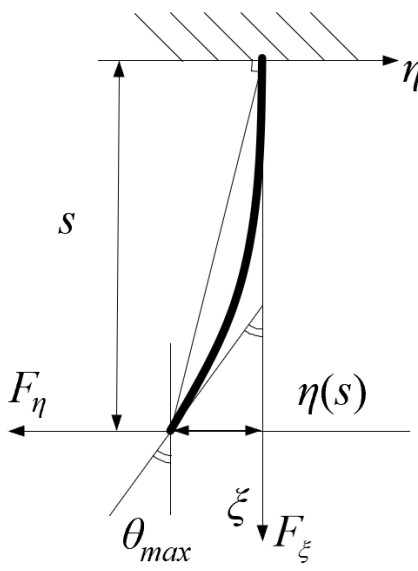

**Figure 2.** Bending diagram of the bristle.

Then, $F_\eta$ can be obtained as $F_\eta = \sqrt{15}EI\sqrt{(L-s)/s^5}$. The relationship between the normal force and the bristle length can be obtained based on Equations (2) and (4) as

$$N = \frac{EI}{s^2\sin(\theta_0 + \varphi + \arctan\mu)}\sqrt{\frac{15(L-s)}{(1+\mu^2)s}} \tag{5}$$

Substituting $\eta(s) = R\sin\varphi\cos\theta_0 - \alpha\sin\theta_0$ into Equation (5) yields

$$\eta(s) = \frac{F_\eta s^3}{3EI} = \frac{\sqrt{15}}{3}\sqrt{Ls - s^2}. \tag{6}$$

Thus,

$$-6R^2\sin^2\psi + (30R(R+\delta)\cos\theta_0 - 15LR)\sin\psi - 18R(R+\delta)\sin\theta_0\cos\psi = \\ 9(R+\delta)^2\sin^2\theta_0 - 15L(R+\delta)\cos\theta_0 - 15(R+\delta)^2\cos^2\theta_0 - 9R^2 \tag{7}$$

The angle $\psi$ can be obtained by solving the above equation. The deflection angle of the bristle can be obtained from Equation (3). Then, normal force $N$ corresponding to any radius gap $\delta$ can be obtained.

The entire forces of the bristle around the brush seal ring can be obtained by integrating the normal force of a single bristle around the brush seal ring [21]. The consequent nonlinear brush seal forces of the bristles on the rotor are

$$
\begin{aligned}
\overline{F}_x &= A^*\sigma_2[(\mu_1 + 2\mu_2\sigma_3 + 3\mu_3\sigma_3^2 + 4\mu_4\sigma_3^3)e + (0.75\mu_3\sigma_2^2 + \mu_4\sigma_2^2\sigma_3)e^3] \\
&\quad \times (\sin\varphi + \cos\varphi)(\sin\gamma + \mu\cos\gamma), \\
\overline{F}_y &= A^*\sigma_2[(\mu_1 + 2\mu_2\sigma_3 + 3\mu_3\sigma_3^2 + 4\mu_4\sigma_3^3)e + (0.75\mu_3\sigma_2^2 + \mu_4\sigma_2^2\sigma_3)e^3] \\
&\quad \times (\sin\varphi + \cos\varphi)(\cos\gamma - \mu\sin\gamma)
\end{aligned}
\tag{8}
$$

where

$$
\begin{aligned}
&\mu_0 = -2.11271 \times 10^3, \mu_1 = 9.08387 \times 10^3, \mu_2 = -1.46323 \times 10^4, \\
&\mu_3 = 1.04705 \times 10^4, \mu_4 = 2.80927 \times 10^3; \\
&A^* = \frac{n}{2L^2}\frac{\sqrt{15EI}}{\sqrt{1+\mu^2}\sin(\theta_0 + \varphi + \arctan\mu)}, \\
&\sigma_1 = R\cos^2\theta_0 - R\cos(\varphi + \theta_0)\cos\theta_0, e = \sqrt{x^2 + y^2}, \\
&\sigma_2 = \cos\theta_0/L, \sigma_3 = (c\cos\theta_0 + \sigma_1)/L.
\end{aligned}
\tag{9}
$$

## 3. Nonlinear Motions of the Brush Seal Rotor System

In this research, we explore the nonlinear motions and bifurcations of a brush seal rotor system. We consider the nondimensional variables as $x = X/Y_s$, $y = Y/Y_s$, $t = \tau/\sqrt{m/k}$, $\alpha = C_i/\sqrt{Mk}$, $\Omega = \omega\sqrt{M/k}$ and illustrate the results in semi-analytical Poincare maps to provide a good understanding of the nonlinear motions and the bifurcations of the brush seal rotor systems.

### 3.1. Bifurcation Characteristics

We apply a discrete mapping method [19,20] to the brush seal rotor system for the bifurcations and nonlinear motions. The discrete mapping method employs implicit mapping structures for nonlinear motion solutions, which are so called semi-analytical solutions. The Jacobian matrix for the bifurcation analysis can be obtained as follows.

$$
\begin{aligned}
\boldsymbol{DP} &= \left[\frac{\partial \mathbf{x}_N}{\partial \mathbf{x}_0}\right]_{(\mathbf{z}_0^*, \mathbf{z}_0^*, \cdots, \mathbf{z}_N^*)} = DP_N \cdot DP_{N-1} \cdot \ldots \cdot DP_2 \cdot DP_1 \\
&= \prod_{k=N}^{1} DP_k
\end{aligned}
\tag{10}
$$

where

$$
DP_k = \left[\frac{\partial \mathbf{x}_k}{\partial \mathbf{x}_{k-1}}\right]_{(\mathbf{x}_k^*, \mathbf{x}_{k-1}^*)} =
\begin{bmatrix}
\frac{\partial x_{1,k}}{\partial x_{1,k-1}} & \frac{\partial x_{1,k}}{\partial x_{2,k-1}} & \frac{\partial x_{1,k}}{\partial y_{1,k-1}} & \frac{\partial x_{1,k}}{\partial y_{2,k-1}} \\
\frac{\partial x_{2,k}}{\partial x_{1,k-1}} & \frac{\partial x_{2,k}}{\partial x_{2,k-1}} & \frac{\partial x_{2,k}}{\partial y_{1,k-1}} & \frac{\partial x_{2,k}}{\partial y_{2,k-1}} \\
\frac{\partial y_{1,k}}{\partial x_{1,k-1}} & \frac{\partial y_{1,k}}{\partial x_{2,k-1}} & \frac{\partial y_{1,k}}{\partial y_{1,k-1}} & \frac{\partial y_{1,k}}{\partial y_{2,k-1}} \\
\frac{\partial y_{2,k}}{\partial x_{1,k-1}} & \frac{\partial y_{2,k}}{\partial x_{2,k-1}} & \frac{\partial y_{2,k}}{\partial y_{1,k-1}} & \frac{\partial y_{2,k}}{\partial y_{2,k-1}}
\end{bmatrix}_{(\mathbf{x}_k^*, \mathbf{x}_{k-1}^*)}
\tag{11}
$$

$x_{1,k}$ and $x_{2,k}(k = 0, 1, 2, \cdots, +\infty)$ are the discrete node points of the displacement and velocity in the $x$-direction. And $y_{1,k}$ and $y_{2,k}(k = 0, 1, 2, \cdots, +\infty)$ are the node points of the

displacement and velocity in the *y*-direction. The detailed components of the above Jacobian matrix have the following expressions. The initial derivation is put in the Appendix A.

$$
\begin{aligned}
&\frac{\partial x_{1,k}}{\partial x_{1,k-1}} = 1 + \frac{1}{2}h\frac{\partial x_{2,k}}{\partial x_{1,k-1}}, \quad \frac{\partial x_{2,k}}{\partial x_{1,k-1}} = \frac{(\Delta_{11}+\Delta_{13})\Delta_{41}+\frac{1}{2}h(\Delta_{22}+\Delta_{23})\Delta_{12}}{\Delta_{31}\Delta_{41}-\frac{1}{4}h^2\Delta_{12}\Delta_{22}}, \\
&\frac{\partial x_{1,k}}{\partial x_{2,k-1}} = \frac{1}{2}h\left(\frac{\partial x_{2,k}}{\partial x_{2,k-1}}+1\right), \quad \frac{\partial x_{2,k}}{\partial x_{2,k-1}} = \frac{(\frac{1}{h}-\frac{1}{2}\alpha+\frac{1}{2}h\Delta_{11})\Delta_{41}+\frac{1}{4}h^2\Delta_{12}\Delta_{22}}{\Delta_{31}\Delta_{41}-\frac{1}{4}h^2\Delta_{12}\Delta_{22}}, \\
&\frac{\partial x_{1,k}}{\partial y_{1,k-1}} = \frac{1}{2}h\frac{\partial x_{2,k}}{\partial y_{1,k-1}}, \quad \frac{\partial x_{2,k}}{\partial y_{1,k-1}} = \frac{(\Delta_{12}+\Delta_{14})\Delta_{41}+\frac{1}{2}h\Delta_{12}(\Delta_{21}+\Delta_{22})}{\Delta_{31}\Delta_{41}-\frac{1}{4}h^2\Delta_{12}\Delta_{22}}, \\
&\frac{\partial x_{1,k}}{\partial y_{2,k-1}} = \frac{1}{2}h\frac{\partial x_{2,k}}{\partial y_{2,k-1}}, \quad \frac{\partial x_{2,k}}{\partial y_{2,k-1}} = \frac{\frac{1}{2}h\Delta_{12}\Delta_{41}+\frac{1}{2}h\Delta_{12}(\frac{1}{h}-\frac{1}{2}\alpha+\frac{1}{2}h\Delta_{21})}{\Delta_{31}\Delta_{41}-\frac{1}{4}h^2\Delta_{12}\Delta_{22}}, a \\
&\frac{\partial y_{1,k}}{\partial x_{1,k-1}} = \frac{1}{2}h\frac{\partial y_{2,k}}{\partial x_{1,k-1}}, \quad \frac{\partial y_{2,k}}{\partial x_{1,k-1}} = \frac{(\Delta_{22}+\Delta_{23})\Delta_{31}+\frac{1}{2}h(\Delta_{11}+\Delta_{13})\Delta_{22}}{\Delta_{31}\Delta_{41}-\frac{1}{4}h^2\Delta_{12}\Delta_{22}}, \\
&\frac{\partial y_{1,k}}{\partial x_{2,k-1}} = \frac{1}{2}h\frac{\partial y_{2,k}}{\partial x_{2,k-1}}, \quad \frac{\partial y_{2,k}}{\partial x_{2,k-1}} = \frac{\frac{1}{2}h\Delta_{22}\Delta_{31}+\frac{1}{2}h\Delta_{22}(\frac{1}{h}-\frac{1}{2}\alpha+\frac{1}{2}h\Delta_{11})}{\Delta_{31}\Delta_{41}-\frac{1}{4}h^2\Delta_{12}\Delta_{22}}, \\
&\frac{\partial y_{1,k}}{\partial y_{1,k-1}} = 1 + \frac{1}{2}h\frac{\partial y_{2,k}}{\partial y_{1,k-1}}, \quad \frac{\partial y_{2,k}}{\partial y_{1,k-1}} = \frac{(\Delta_{21}+\Delta_{24})\Delta_{31}+\frac{1}{2}h\Delta_{22}(\Delta_{12}+\Delta_{14})}{\Delta_{31}\Delta_{41}-\frac{1}{4}h^2\Delta_{12}\Delta_{22}}, \\
&\frac{\partial y_{2,k}}{\partial y_{2,k-1}} = \frac{1}{2}h\left(\frac{\partial y_{2,k}}{\partial y_{2,k-1}}+1\right), \quad \frac{\partial y_{2,k}}{\partial y_{2,k-1}} = \frac{(\frac{1}{h}-\frac{1}{2}\alpha+\frac{1}{2}h\Delta_{21})\Delta_{31}+\frac{1}{4}h^2\Delta_{12}\Delta_{22}}{\Delta_{31}\Delta_{41}-\frac{1}{4}h^2\Delta_{12}\Delta_{22}}.
\end{aligned} \tag{12}
$$

with

$$
\begin{aligned}
\Delta_{11} &= -\frac{1}{2}\beta_1 + \frac{3}{2}\eta x_{1km}^2 - \mu\eta x_{1km}y_{1km} + \frac{1}{2}\eta y_{1km}^2, \\
\Delta_{12} &= -\frac{1}{2}\gamma - \frac{1}{2}\mu\eta x_{1km}^2 + \eta x_{1km}y_{1km} - \frac{3}{2}\mu\eta y_{1km}^2, \\
\Delta_{13} &= -\frac{1}{2}\beta_1 + \frac{3}{2}\eta x_{1km} - \mu\eta x_{1km}y_{1km} + \frac{1}{2}\eta y_{1km}^2, \\
\Delta_{14} &= -\frac{1}{2}\gamma - \frac{1}{2}\mu\eta x_{1km}^2 + \eta x_{1km}y_{1km} - \frac{3}{2}\mu\eta y_{1km}^2 \\
\Delta_{21} &= -\frac{1}{2}\beta_2 + \frac{3}{2}\eta y_{1km}^2 + \mu\eta x_{1km}y_{1km} + \frac{1}{2}\eta x_{1km}^2, \\
\Delta_{22} &= -\frac{1}{2}\gamma + \frac{1}{2}\mu\eta y_{1km}^2 + \eta x_{1km}y_{1km} + \frac{3}{2}\mu\eta x_{1km}^2, \\
\Delta_{23} &= -\frac{1}{2}\gamma + \frac{1}{2}\mu\eta y_{1km}^2 + \eta x_{1km}y_{1km} + \frac{3}{2}\mu\eta x_{1km}^2 \\
\Delta_{24} &= -\frac{1}{2}\beta_2 + \frac{3}{2}\eta y_{1km}^2 + \mu\eta x_{1km}y_{1km} + \frac{1}{2}\eta x_{1km}^2, \\
\Delta_{31} &= \left[\frac{1}{h}+\frac{1}{2}\alpha - \frac{1}{2}h\left(-\frac{1}{2}\beta_1+\frac{3}{2}\eta x_{1km}^2 - \mu\eta x_{1km}y_{1km}+\frac{1}{2}\eta y_{1km}^2\right)\right], \\
\Delta_{41} &= \left[\frac{1}{h}+\frac{1}{2}\alpha - \frac{1}{2}h\left(-\frac{1}{2}\beta_2+\frac{3}{2}\eta y_{1km}^2 + \mu\eta x_{1km}y_{1km}+\frac{1}{2}\eta x_{1km}^2\right)\right].
\end{aligned} \tag{13}
$$

where $x_{1km}$, $x_{2km}$, $y_{1km}$, and $y_{2km}$ are the middle points of the corresponding discrete nodes on the displacement and velocity orbits, whose expressions can be given as

$$
\begin{aligned}
x_{1km} &= \frac{1}{2}(x_{1,k}+x_{1,k-1}), \quad y_{1km} = \frac{1}{2}(y_{1,k}+y_{1,k-1}), \\
x_{2km} &= \frac{1}{2}(x_{2,k}+x_{2,k-1}), \quad y_{2km} = \frac{1}{2}(y_{2,k}+y_{2,k-1}).
\end{aligned}
$$

The detailed discretization procedures can be found in [19,20] so they are skipped in this part. In the following parts, "SN", "NB", and "PD" represent the saddle node, Neimark, and period-doubling bifurcations. Solid lines represent stable nonlinear motions and dashed lines represent unstable nonlinear motions. "U" means unstable bifurcations or motions.

### 3.2. Numerical Simulation

For comparison of the nonlinear motions, numerical simulation is performed to compare the semi-analytical and the numerical results. The numerical results can be obtained by some numerical integration method such as the Runge–Kutta method or Euler method. This paper adopts an implicit mid-point method [20] for numerical results. The solid lines

represent the numerical results, while the circles represent the semi-analytical motions. I.C. means the initial conditions.

Figure 3 shows the semi-analytical motions and numerical results of the stable period-1 motion. It can be seen from Figure 3a,b that the displacements are simple sinusoidal curves. The phase diagrams are simple ellipses in Figure 3c,d. The semi-analytical motions of the period-1 motion are compared with complete agreement with the numerical results. The nonlinear rotor system needs two more harmonic terms to meet the engineering requirements if the accuracy required is $\varepsilon < 10^{-3}$ in Figure 3e,f. For the accuracy of $\varepsilon < 10^{-9}$, three harmonic terms are required to meet the theoretical design or control requirements.

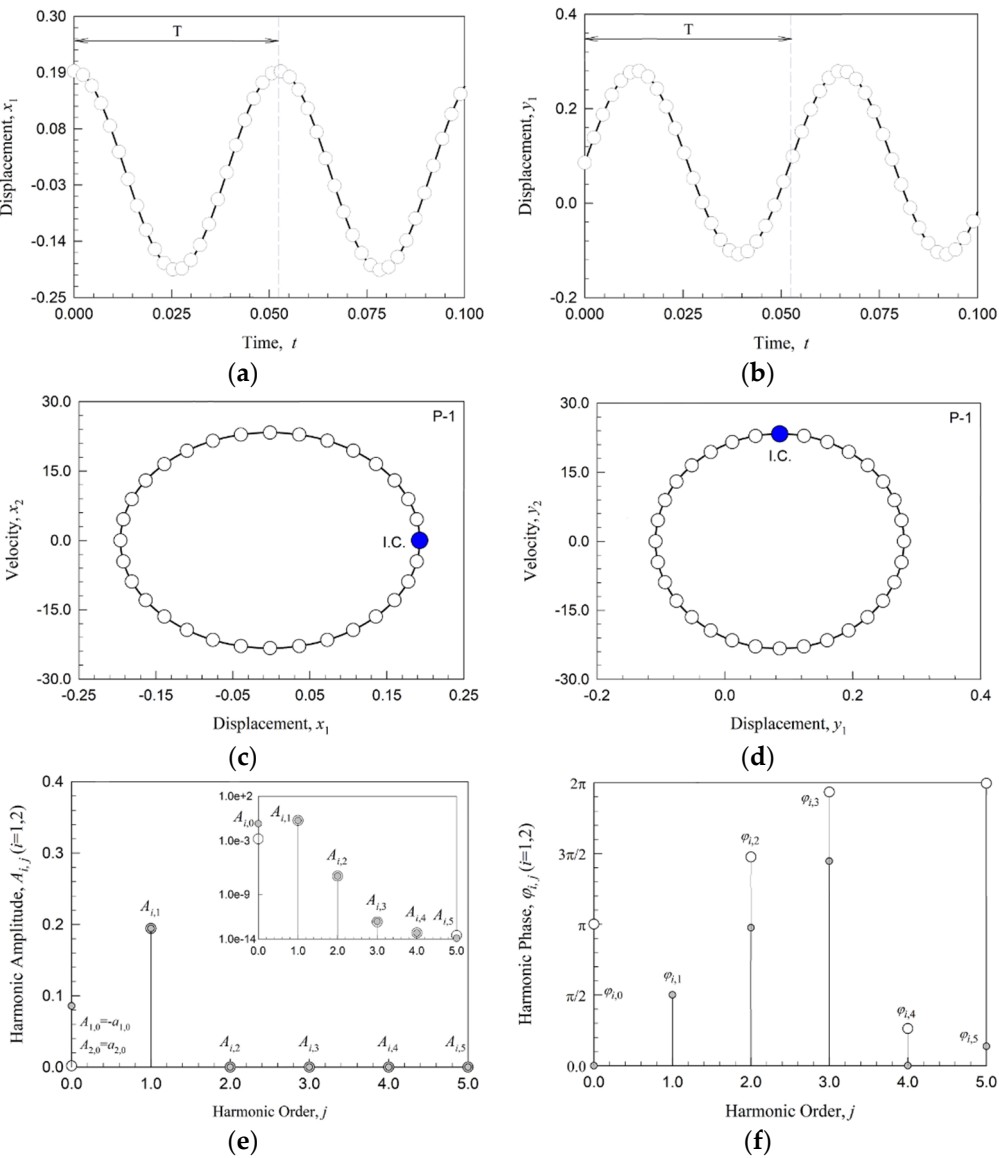

**Figure 3.** Numerical comparison of stable period-1 motion of brush seal rotor system: (**a**) displacement, $x_1$; (**b**) displacement, $y_1$; (**c**) phase diagram, $(x_1, x_2)$; (**d**) phase diagram, $(y_1, y_2)$; (**e**) harmonic spectrum, $A_{i,j}$; (**f**) harmonic phases, $\varphi_{i,j}$. ($i = 1$ and 2 for *x*- and *y*-directions, respectively; $j = 0, 1, 2, \cdots, 5$).

Figure 4 shows the comparison of the unstable period-1 motion of the brush seal rotor system. In Figure 4a,b, if the semi-analytical motions and the numerical results are set at the same initial conditions, the numerical motions are consistent with the analytic results in $t = [0, 1.5]$. But when t > 1.5, the numerical results gradually move away from the semi-analytical results and the dangerous motion happens. This confirms the unstable

motions. Figure 4c,d present the same unstable phenomenon. From Figure 4e,f, if the accuracy required is set for $\varepsilon < 10^{-3}$, the brush seal rotor system needs two harmonic terms to meet the engineering requirements. For accuracy $\varepsilon < 10^{-10}$, five harmonic terms are required to meet the theoretical design or control requirements.

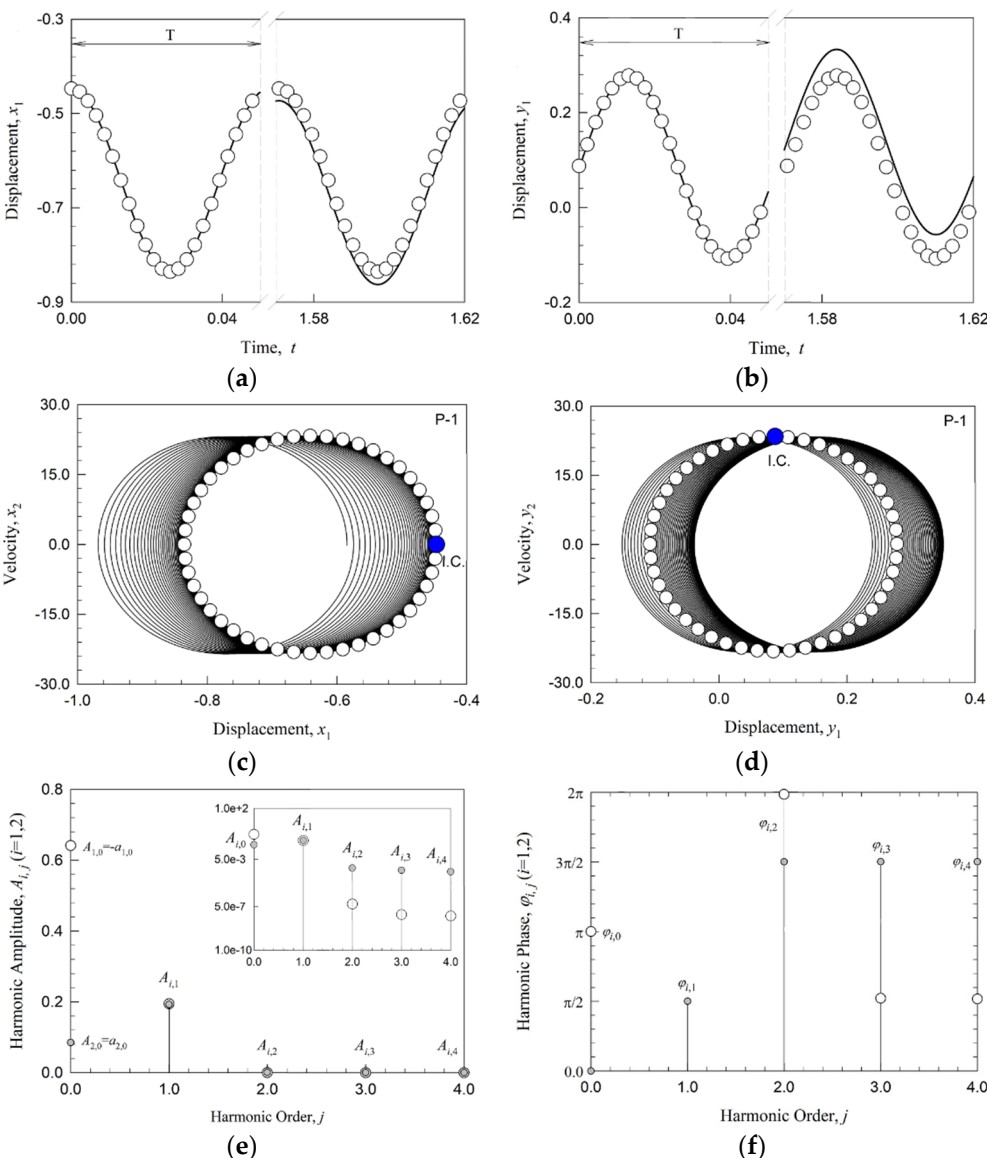

**Figure 4.** Numerical comparison of unstable period-1 motion of brush seal rotor system: (**a**) displacement, $x_1$; (**b**) displacement, $y_1$; (**c**) phase diagram, $(x_1, x_2)$; (**d**) phase diagram, $(y_1, y_2)$; (**e**) harmonic spectrum, $A_{i,j}$; (**f**) harmonic phases, $\varphi_{i,j}$. ($i = 1$ and 2 for $x$- and $y$-directions, respectively; $j = 0, 1, 2, \cdots, 4$).

Figure 5 presents the stable period-2 motion of the brush seal rotor system at $\Omega = 10.2$. In Figure 5a,b, the numerical motions in solid lines match with the semi-analytical solutions in circles. This confirms stable period-2 motion solutions. Figure 5c,d present the displacement orbit and velocity orbit. The exact match of the numerical motions with the semi-analytical solutions validates the correctness of the obtained semi analytical motions. Figure 5e,f present the harmonic amplitudes and phases of the stable period-2 motion. For displacement $x_1$, the most contributed harmonic amplitudes are $A_{1,0} = 0.1027$, $A_{1,0.5} = 0.1774$, $A_{1,1} = 0.4730$, and $A_{1,1.5} = 0.0136$. For displacement $y_1$, the most contributed harmonic amplitudes are $A_{2,0} = 0.2007$, $A_{2,0.5} = 0.8378$, $A_{2,1} = 0.5680$, $A_{2,1.5} = 0.0367$, and

$A_{2,2} = 0.0129$. The brush seal rotor system needs twenty harmonic terms to meet the engineering requirements of the accuracy required, set for $\varepsilon < 10^{-12}$.

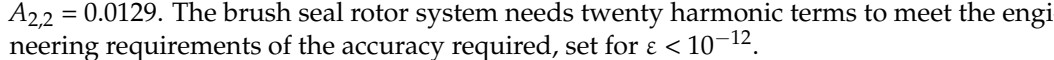

**Figure 5.** Numerical comparison of the stable period-2 motion of the brush seal rotor system: (**a**) displacement, $x_1$; (**b**) displacement, $y_1$; (**c**) displacement plane, $(x_1, y_1)$; (**d**) velocity plane, $(x_2, y_2)$; (**e**) harmonic spectrum, $A_{i,j}$; (**f**) harmonic phases, $\varphi_{i,j}$. ($i = 1$ and 2 for $x$- and $y$-directions, respectively; $j = 0, 1, 2, \cdots, 10$).

Figure 6 presents the unstable period-2 motion of the brush seal rotor system at $\Omega = 7.38$. In Figure 6a,b, the numerical solutions in solid lines almost move along with the semi-analytical motions in circles. Even the period-2 motion is unstable. The actual nonlinear motion does not cause a dangerous motion in short time. Figure 6c,d present the displacement orbit and velocity orbit. The unstable period-2 motions all follow the unstable semi-analytical solutions, and the unstable motion turns very steady. Figure 6e,f present the harmonic amplitudes and phases of the unstable period-2 motion. The most contributed harmonic amplitudes for unstable semi-analytical displacement $x_1$ are $A_{1,0} = 0.0609$, $A_{1,0.5} = 0.3276$, $A_{1,1} = 0.5478$, and $A_{1,1.5} = 0.0153$. The most contributed harmonic amplitudes for unstable semi-analytical displacement $y_1$ are $A_{2,0} = 0.4913$, $A_{2,0.5} = 0.2155$, $A_{2,1} = 0.6592$,

$A_{2,1.5}$ = 0.0148 and $A_{2,2}$ = 0.0160. The most contributed harmonics can be used for control of such unstable period-2 motion to keep the original nonlinearity of the motions, while less harmonics are needed to reach the engineering requirements.

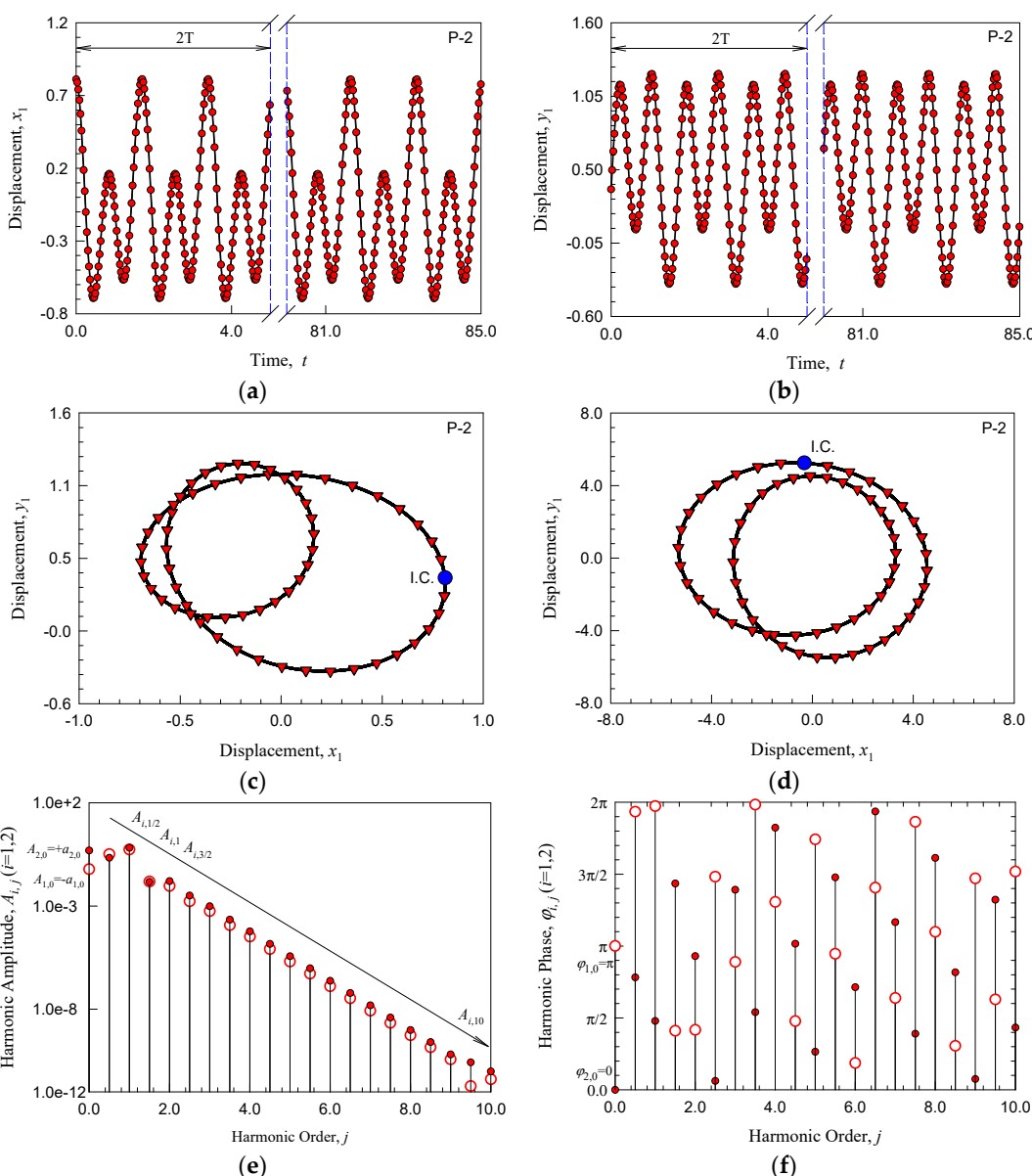

**Figure 6.** Numerical comparison of unstable period-2 motion of brush seal rotor system: (**a**) displacement, $x_1$; (**b**) displacement, $y_1$; (**c**) displacement plane, $(x_1, y_1)$; (**d**) velocity plane, $(x_2, y_2)$; (**e**) harmonic spectrum, $A_{i,j}$; (**f**) harmonic phases, $\varphi_{i,j}$. ($i$ = 1 and 2 for $x$- and $y$-directions, respectively; $j = 0, 1, 2, \cdots, 10$).

### 3.3. Bifurcation Diagrams

For a better understanding of stable and unstable motion switching, the nonlinear motions and bifurcations are projected in 2D planes in Figure 7. The solid lines mean stable motions and the dashed lines mean unstable motions. Between stable and unstable motions, bifurcations happen. The stable and unstable displacement $x_1$ is depicted by Figure 7a. When the mass eccentricity is set to be e = 0.0005, the stable displacement experiences a jumping at $\Omega$ = 1.0955 because of a saddle node bifurcation where a nonlinear resonance happens. The resonance is more obvious at the $y$-direction in Figure 7c. The dimensionless rotor system is characterized with a linear resonance frequency of $\Omega'$ = 1. For the original

system, the linear resonance frequency is around 1027 r/min. Because of the nonlinearity of the brush seal forces, the resonance frequency will shift. The nonlinear resonance of the rotor system increases with the increase in the mass eccentricity. So do the resonance frequencies. The motion jumps at speed $\Omega = 1.1787$ when the eccentricity grows to $e = 0.0015$ where a saddle node bifurcation occurs. Because of saddle node bifurcations, the nonlinear rotor system may experience chatter and non-smooth vibration during running. When the rotor runs to $\Omega = 2.0500$, the stable period-1 motion changes to stable period-2 motion because of a period-doubling bifurcation. When the rotational speed increases to $\Omega = 2.0600$, a period-doubling bifurcation occurs and the stable period-2 motion disappears.

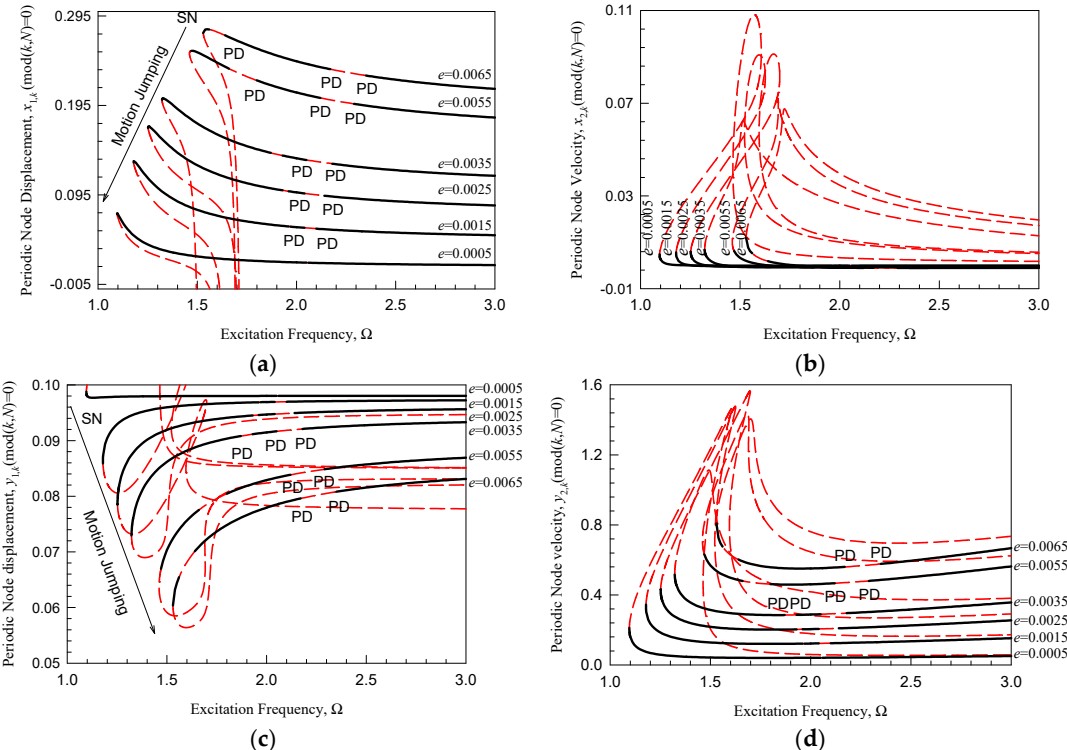

**Figure 7.** Semi-analytical Poincare map of the nonlinear period-1 motions in the rotor system with various eccentricities: (**a**) Poincare displacement, $x_{1,k}$; (**b**) Poincare velocity, $x_{2,k}$; (**c**) Poincare displacement, $y_{1,k}$; (**d**) Poincare velocity, $y_{2,k}$. ($k = 0, 1, 2, \cdots, +\infty$).

The bifurcations on the velocity in the $x$-direction of the nonlinear rotor system are illustrated in Figure 7b. On the stable motion, there is an increase in the magnitude of the velocity along the $x$-direction. It is important to note that the velocity and the displacement in the $x$-direction have the same stable and unstable speed ranges, as well as the bifurcations. When the speed is operated to much lower than the critical speed, the nonlinear motion of the rotor remains steady regardless of the mass eccentricities. The vibration velocities approach almost zeros. The period-doubling bifurcations are $\Omega = \{1.98, 2.05, 2.06\}$ for $e = 0.0005$, $\Omega = \{1.96, 1.97, 2.04, 2.1\}$ for $e = 0.0015$; $\Omega = \{1.93, 1.96, 2.04, 2.14\}$ for $e = 0.0025$; $\Omega = \{1.87, 1.93, 2.05, 2.18\}$ for $e = 0.0035$; $\Omega = \{1.66, 1.8, 2.12, 2.29\}$ for $e = 0.0055$; $\Omega = \{1.56, 1.64, 2.17, 2.35\}$ for $e = 0.0065$.

The displacement $y_{1,k}$ of the nonlinear motions in the brush seal rotor system is presented in Figure 7c. At the location where the saddle node bifurcation is located, the displacement chatters in the $y$-direction. Figure 7d illustrates the velocity $y_{2,k}$ varying with the rotational speed in the $y$-direction. The nonlinear phenomenon of $y_{2,k}$ is similar to that in the $x$-direction.

For obtaining the period-2 motions, the eigenvalue analysis of the brush seal rotor system is performed in Figure 8. The solid lines mean the stable motions and the dashed lines mean unstable motions. The nonlinear motion interacts with stable and unstable

branches. For clear observation, the eigenvalue analysis and the bifurcations are illustrated in two separate speed ranges. The eigenvalue analysis within $\Omega \in (5.6, 10.1)$ is plotted in Figure 8a,c,e. Because the nonlinear motions within $\Omega \in (0.0, 5.6)$ are all stable, the eigenvalues within this speed range are all with magnitudes less than one, so this part is skipped. During the increasing speed process, a period-doubling bifurcation occurs at $\Omega = 9.23$ with one eigenvalue greater than one, one eigenvalue crossing the negative-one line, and the others smaller than one so that unstable period-2 motions may occur with small perturbation. Such unstable period-2 motions come from unstable period-1 motions. The nonlinear rotor usually accompanies dangerous motions with large displacement. A stable period-doubling bifurcation happens at $\Omega = 7.85$, where period-4 motions will happen. The eigenvalue analysis and bifurcations within $\Omega \in (10.0, 15.0)$ are presented in Figure 8b,d,f. Two period-doubling bifurcations are discovered at $\Omega = 11.47$ and $\Omega = 12.32$ where period-2 and 4 motions will be produced during running. The real, imaginary, and magnitude parts of the eigenvalues provide a quantitative understanding of the bifurcations in the brush seal rotor system.

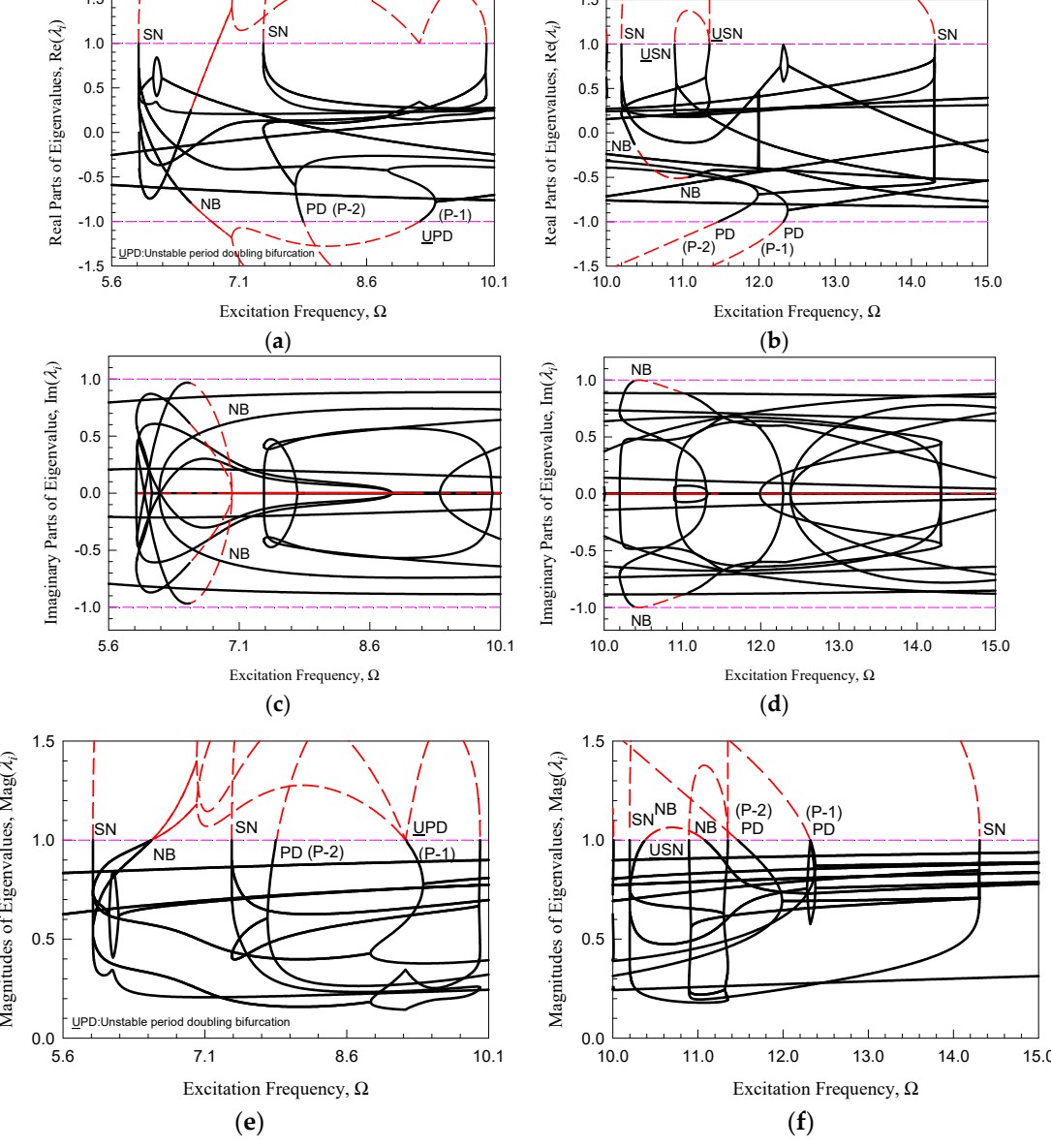

**Figure 8.** Bifurcation analysis by eigenvalues. I. $\Omega \in (5.6, 10.1)$: (**a**) real parts, Re$(\lambda_i)$; (**c**) imaginary parts, Im$(\lambda_i)$; (**e**) magnitudes, Mag$(\lambda_i)$; II. $\Omega \in (10.0, 15.0)$: (**b**) real parts, Re$(\lambda_i)$; (**d**) imaginary parts, Im$(\lambda_i)$; (**f**) magnitudes, Mag$(\lambda_i)$.

In Figures 9 and 10, the semi-analytical Poincare maps of period-2 motions are presented to illustrate the bifurcations and nonlinear motions. The solid lines mean stable motions and the dashed lines mean unstable motions. When the speed is increasing, the period-1 motion is stable. A saddle node bifurcation happens at $\Omega = 5.917$ to generate chatting during the speed-increasing process. A Neimark bifurcation happens at $\Omega = 6.53$ and an unstable quasi-period-1 motion happens. The period-doubling bifurcation at $\Omega = 9.23$ causes a period-2 motion, and half-frequency components in the nonlinear motions emerge. Another period-doubling bifurcation at $\Omega = 7.85$ causes a period-4 motion, and quarter-frequency components in the nonlinear motions emerge. Moreover, quasi-periodic period-2 motions are produced at Neimark bifurcations on $\Omega = 10.37$ and $\Omega = 11.08$. Nonlinear chattering and jumping motions are produced at saddle node bifurcations at $\Omega = 10.197$ and $\Omega = 14.307$. Period-2 and period-4 motions are produced at period-doubling bifurcations on $\Omega = 11.47$ and $\Omega = 12.32$ in the brush seal rotor system. The nonlinear motions behave stably and unstably as well as at complex bifurcations. Attention should be paid to the practical rotor design to put the working speed range smaller than these stability shifts. Care must be taken in operation to avoid unstable and dangerous nonlinear motions.

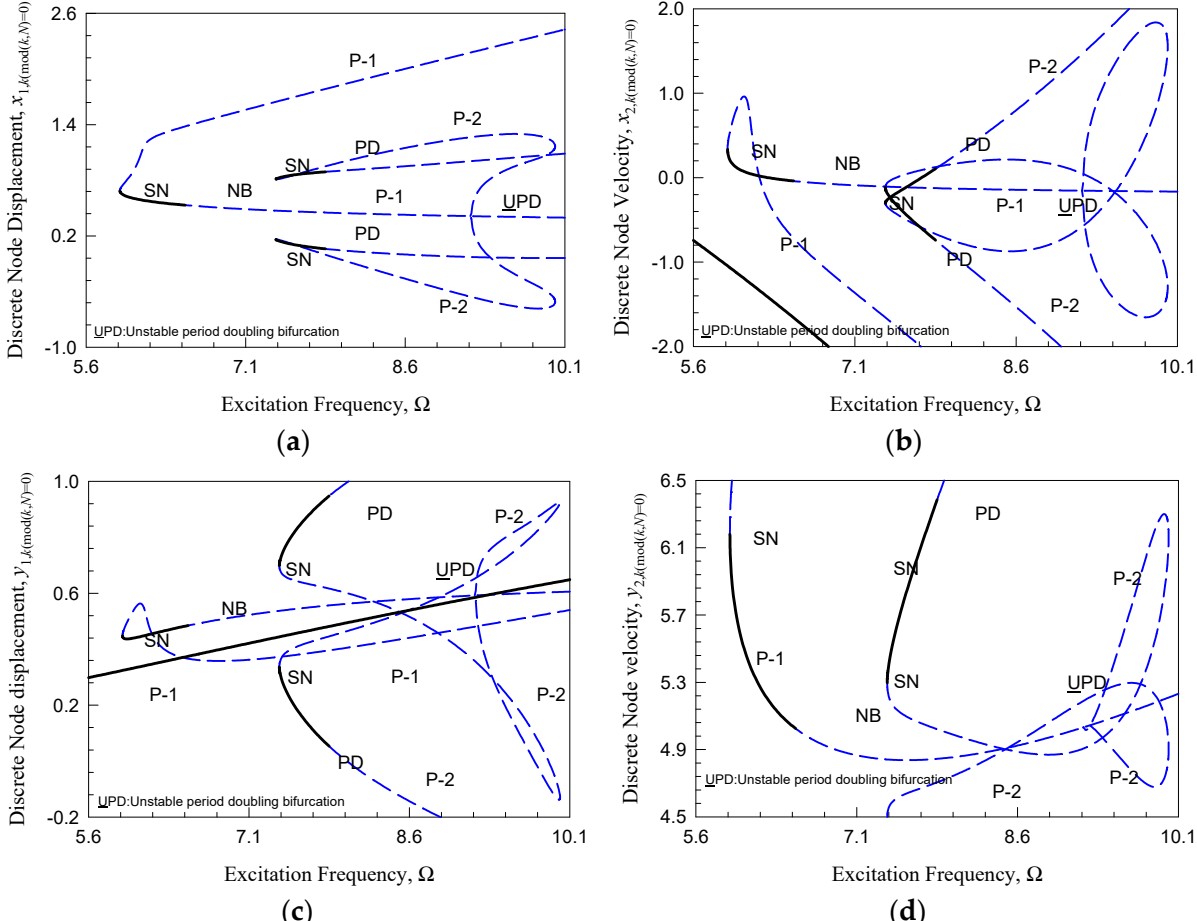

**Figure 9.** Semi-analytical Poincare map of the nonlinear period-2 motions in the rotor system within $\Omega \in (5.6, 10.1)$: (**a**) Poincare displacement, $x_{1,k}$; (**b**) Poincare velocity, $x_{2,k}$; (**c**) Poincare displacement, $y_{1,k}$; (**d**) Poincare velocity, $y_{2,k}$. ($k = 0, 1, 2, \cdots ,$).

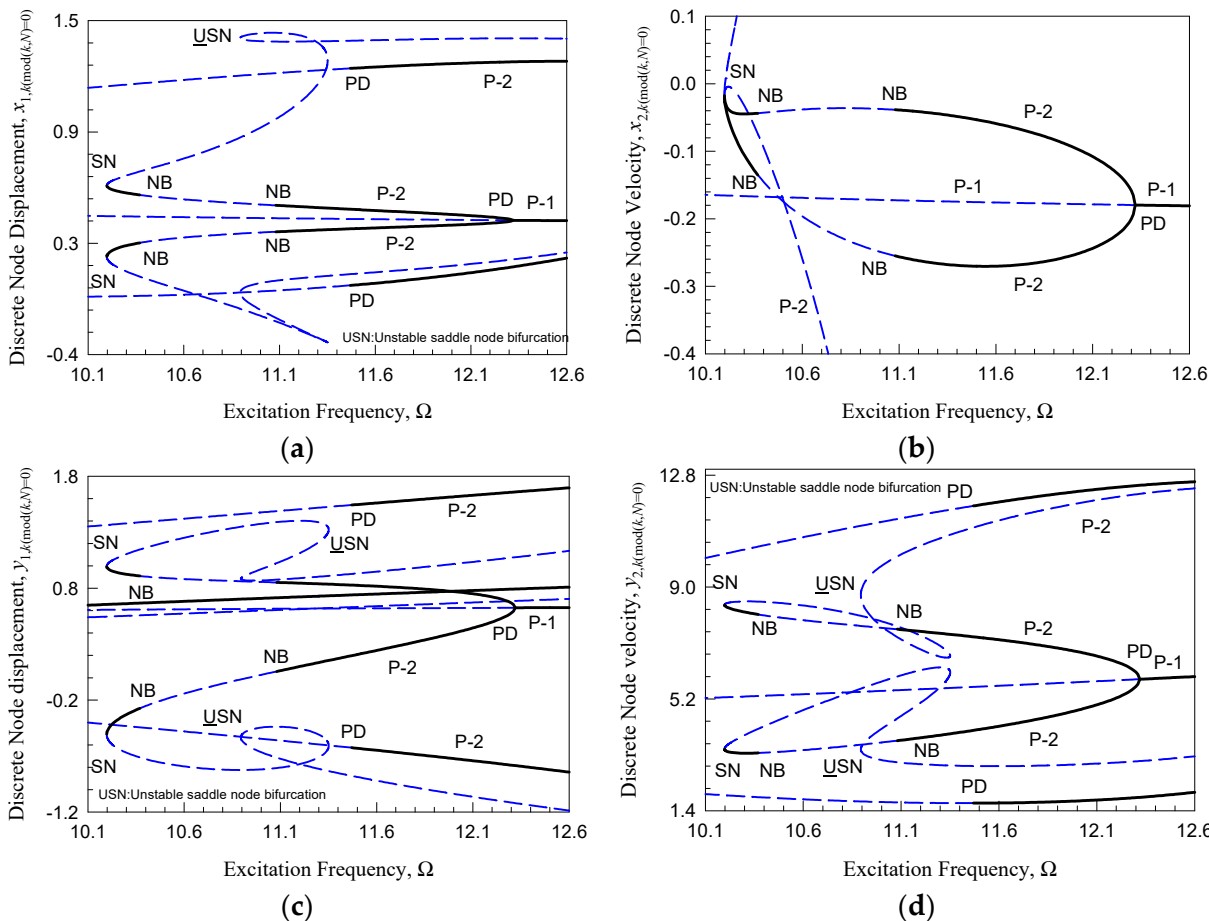

**Figure 10.** Semi-analytical Poincare map of the nonlinear period-2 motions in the rotor system within $\Omega \in (10.1, 12.6)$: (**a**) Poincare displacement, $x_{1,k}$; (**b**) Poincare velocity, $x_{2,k}$; (**c**) Poincare displacement, $y_{1,k}$; (**d**) Poincare velocity, $y_{2,k}$. ($k = 0, 1, 2, \cdots,$).

### 3.4. Parameter Maps of SN and PD

Since the saddle node and period-doubling bifurcation have the most influence in the nonlinear motions of the brush seal rotor system, Figure 11 depicts parameter maps for stable and unstable domains of saddle node and period-doubling bifurcations. Figure 11a displays the parameter maps of the period-doubling bifurcations of nonlinear motions. The maps for unstable nonlinear motions are depicted by the blue and black, while the stable motion domain is depicted by green. The boundaries are period-doubling bifurcations for producing period-2 motions. On the boundaries, the period-1 motions are turned to period-2 motions. Under such parameters, the left unstable domain disappears at $\Omega = 2.0520$ with $e = 0.0024$. The lower left period-doubling bifurcation domain disappears at $\Omega = 1.9809$ with $e = 0.00494$. For the right side, the unstable domain increases as the rotational speed increases.

The parameter map for the saddle node bifurcations is shown in Figure 11b. The purple represents the unstable map while the green represents the stable map. The saddle node bifurcations are located on the boundaries between the stable and unstable maps. On such boundaries, the nonlinear motions of the brush seal rotor are characterized with a chattering phenomenon. Safe operation should be considered to avoid damage to the rotor system.

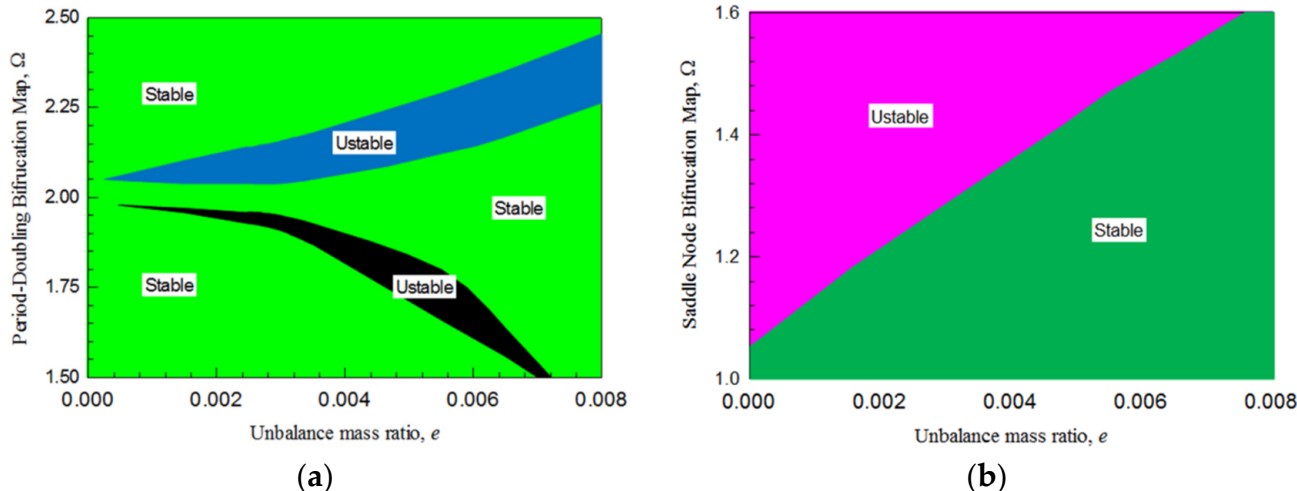

**Figure 11.** Parameter maps of bifurcations in the brush seal rotor system: (**a**) parameter map of period-doubling bifurcations; (**b**) parameter map of saddle node bifurcations.

## 4. Conclusions

In this research, the elasto-dynamic theory was utilized to model brush seal forces. A nonlinear brush seal rotor system was constructed by combination of a flexible rotor system with nonlinear brush seal forces. The nonlinear motions were obtained by a semi-analytical approach as well as stability and bifurcations. The system chattered nonlinearly at specific speeds near saddle node bifurcations. Parameter maps of bifurcations and stability domains of the brush seal rotor system were also achieved. Complicated vibrations and bifurcations were successfully obtained by eigenvalues and exhibited in semi-analytical Poincare maps. Period-doubling bifurcations were obtained to lead the route from period-1 motion to period-2 motion. Numerical comparisons were performed to verify the theoretical results. The period-doubling bifurcations caused motion switch and subharmonic-1/2 vibrations in the brush seal rotor system. The saddle node bifurcations on the parameter map make it possible for the system to prevent nonlinear motion jumping and reduce the harmful motions if optimal parameters are considered. Nonlinearity is an avoidable effect in rotor system design and production. With the increase in the complexity of the practical rotor system, this research provides a method to solve nonlinear resonance, stability, and bifurcations, which will be helpful in future designs to avoid dangerous vibrations.

**Author Contributions:** Methodology, Y.X.; Validation, Y.Z. (Yongde Zhang); Investigation, Y.Z. (Yingyong Zou); Writing—review & editing, Z.X.; Supervision, Y.X.; Funding acquisition, M.W. and D.L. All authors have read and agreed to the published version of the manuscript.

**Funding:** This work is supported by the Jilin Provincial Natural Science Foundation (Grant No. 20230101208JC), the Key R&D and Transformation Plan Project of Qinghai Province (No. 2023-QY-215), National Nature Science Foundation of China (Grant No. 12102319), and the Fundamental Research Funds for the Central Universities (Grant No. xzy012021004).

**Data Availability Statement:** The datasets generated during and/or analyzed during the current study are available from the corresponding author on reasonable request.

**Conflicts of Interest:** The authors declare they have no conflict of interest.

## Appendix A

The initial derivation of the components in the Jacobian matrix has the following forms.

$$\frac{\partial x_{1,k}}{\partial x_{1,k-1}} = 1 + \frac{1}{2}h\frac{\partial x_{2,k}}{\partial x_{1,k-1}}, \frac{\partial x_{1,k}}{\partial x_{2,k-1}} = \frac{1}{2}h(1+\frac{\partial x_{2,k}}{\partial x_{2,k-1}}),$$

$$\frac{\partial x_{1,k}}{\partial y_{1,k-1}} = \frac{1}{2}h\frac{\partial x_{2,k}}{\partial y_{1,k-1}}, \frac{\partial x_{1,k}}{\partial y_{2,k-1}} = \frac{1}{2}h\frac{\partial x_{2,k}}{\partial y_{2,k-1}},$$

$$\frac{\partial x_{2,k}}{\partial x_{1,k-1}} = h\left\{-\frac{1}{2}\alpha\frac{\partial x_{2,k}}{\partial x_{1,k-1}} - \frac{1}{2}\beta_1(\frac{\partial x_{1,k}}{\partial x_{1,k-1}}+1) + \eta[(\frac{3}{2}x_{1km}-\mu x_{1km}y_{1km}+\frac{1}{2}y_{1km}^2)(\frac{\partial x_{1,k}}{\partial x_{1,k-1}}+1)\right.$$
$$\left. -(\frac{1}{2}\mu x_{1km}^2-x_{1km}y_{1km}+\frac{3}{2}\mu y_{1km}^2)\frac{\partial y_{1,k}}{\partial x_{1,k-1}}] - \frac{1}{2}\gamma\frac{\partial y_{1,k}}{\partial x_{1,k-1}}\right\},$$

$$\frac{\partial x_{2,k}}{\partial x_{2,k-1}} = 1 + h\left\{-\frac{1}{2}\alpha(\frac{\partial x_{2,k}}{\partial x_{2,k-1}}+1) - \frac{1}{2}\beta_1\frac{\partial x_{1,k}}{\partial x_{2,k-1}} + \eta[(\frac{3}{2}x_{1km}^2-\mu x_{1km}y_{1km}+\frac{1}{2}y_{1km}^2)\frac{\partial x_{1,k}}{\partial x_{2,k-1}}\right.$$
$$\left. -(\frac{1}{2}\mu x_{1km}^2-x_{1km}y_{1km}+\frac{3}{2}\mu y_{1km}^2)\frac{\partial y_{1,k}}{\partial x_{2,k-1}}] - \frac{1}{2}\gamma\frac{\partial y_{1,k}}{\partial x_{2,k-1}}\right\},$$

$$\frac{\partial x_{2,k}}{\partial y_{1,k-1}} = h\left\{-\frac{1}{2}\alpha\frac{\partial x_{2,k}}{\partial y_{1,k-1}} - \frac{1}{2}\beta_1\frac{\partial x_{1,k}}{\partial y_{1,k-1}} + \eta[(\frac{3}{2}x_{1km}^2-\mu x_{1km}y_{1km}+\frac{1}{2}y_{1km}^2)\frac{\partial x_{1,k}}{\partial y_{1,k-1}}\right.$$
$$\left. -(\frac{1}{2}\mu x_{1km}^2-x_{1km}y_{1km}+\frac{3}{2}\mu y_{1km}^2)(\frac{\partial y_{1,k}}{\partial y_{1,k-1}}+1)] - \frac{1}{2}\gamma(\frac{\partial y_{1,k}}{\partial y_{1,k-1}}+1)\right\},$$

$$\frac{\partial x_{2,k}}{\partial y_{2,k-1}} = h\left\{-\frac{1}{2}\alpha\frac{\partial x_{2,k}}{\partial y_{2,k-1}} - \frac{1}{2}\beta_1\frac{\partial x_{1,k}}{\partial y_{2,k-1}} + \eta[(\frac{3}{2}x_{1km}^2-\mu x_{1km}y_{1km}+\frac{1}{2}y_{1km}^2)\frac{\partial x_{1,k}}{\partial y_{2,k-1}}\right.$$
$$\left. -(\frac{1}{2}\mu x_{1km}^2-x_{1km}y_{1km}+\frac{3}{2}\mu y_{1km}^2)\frac{\partial y_{1,k}}{\partial y_{2,k-1}}] - \frac{1}{2}\gamma\frac{\partial y_{1,k}}{\partial y_{2,k-1}}\right\},$$

$$\frac{\partial y_{1,k}}{\partial x_{1,k-1}} = \frac{1}{2}h\frac{\partial y_{2,k}}{\partial y_{1,k-1}}, \frac{\partial y_{1,k}}{\partial x_{2,k-1}} = \frac{1}{2}h\frac{\partial y_{2,k}}{\partial x_{2,k-1}},$$

$$\frac{\partial y_{1,k}}{\partial y_{1,k-1}} = 1 + \frac{1}{2}h\frac{\partial y_{2,k}}{\partial y_{1,k-1}}, \frac{\partial y_{1,k}}{\partial y_{2,k-1}} = \frac{1}{2}h(1+\frac{\partial y_{2,k}}{\partial y_{2,k-1}}),$$

$$\frac{\partial y_{2,k}}{\partial x_{1,k-1}} = h\left\{-\frac{1}{2}\alpha\frac{\partial y_{2,k}}{\partial x_{1,k-1}} - \frac{1}{2}\beta_2\frac{\partial y_{1,k}}{\partial x_{1,k-1}} + \eta[(\frac{3}{2}y_{1km}^2+\mu x_{1km}y_{1km}+\frac{1}{2}x_{1km}^2)\frac{\partial y_{1,k}}{\partial x_{1,k-1}}\right.$$
$$\left. +(\frac{1}{2}\mu y_{1km}^2+x_{1km}y_{1km}+\frac{3}{2}\mu x_{1km}^2)(\frac{\partial x_{1,k}}{\partial x_{1,k-1}}+1)] - \frac{1}{2}\gamma(\frac{\partial x_{1,k}}{\partial x_{1,k-1}}+1)\right\},$$

$$\frac{y_{2,k}}{\partial x_{2,k-1}} = h\left\{-\frac{1}{2}\alpha\frac{\partial y_{2,k}}{\partial x_{2,k-1}} - \frac{1}{2}\beta_2\frac{\partial y_{1,k}}{\partial x_{2,k-1}} + \eta[(\frac{3}{2}y_{1km}^2+\mu x_{1km}y_{1km}+\frac{1}{2}x_{1km}^2)\frac{\partial y_{1,k}}{\partial x_{2,k-1}}\right.$$
$$\left. +(\frac{1}{2}\mu y_{1km}^2+x_{1km}y_{1km}+\frac{3}{2}\mu x_{1km}^2)(\frac{\partial x_{1,k}}{\partial x_{2,k-1}}+1)] - \frac{1}{2}\gamma\frac{\partial x_{1,k}}{\partial x_{2,k-1}}\right\},$$

$$\frac{y_{2,k}}{\partial x_{2,k-1}} = h\left\{-\frac{1}{2}\alpha\frac{\partial y_{2,k}}{\partial x_{2,k-1}} - \frac{1}{2}\beta_2\frac{\partial y_{1,k}}{\partial x_{2,k-1}} + \eta[(\frac{3}{2}y_{1km}^2+\mu x_{1km}y_{1km}+\frac{1}{2}x_{1km}^2)\frac{\partial y_{1,k}}{\partial x_{2,k-1}}\right.$$
$$\left. +(\frac{1}{2}\mu y_{1km}^2+x_{1km}y_{1km}+\frac{3}{2}\mu x_{1km}^2)(\frac{\partial x_{1,k}}{\partial x_{2,k-1}}+1)] - \frac{1}{2}\gamma\frac{\partial x_{1,k}}{\partial x_{2,k-1}}\right\},$$

$$\frac{y_{2,k}}{\partial y_{1,k-1}} = h\left\{-\frac{1}{2}\alpha\frac{\partial y_{2,k}}{\partial y_{1,k-1}} - \frac{1}{2}\beta_2(\frac{\partial y_{1,k}}{\partial y_{1,k-1}}+1) + \eta[(\frac{3}{2}y_{1km}^2+\mu x_{1km}y_{1km}+\frac{1}{2}x_{1km}^2)(\frac{\partial y_{1,k}}{\partial y_{1,k-1}}+1)\right.$$
$$\left. +(\frac{1}{2}\mu y_{1km}^2+x_{1km}y_{1km}+\frac{3}{2}\mu x_{1km}^2)\frac{\partial x_{1,k}}{\partial y_{1,k-1}}] - \frac{1}{2}\gamma\frac{\partial x_{1,k}}{\partial y_{1,k-1}}\right\},$$

$$\frac{y_{2,k}}{\partial y_{2,k-1}} = 1 + h\left\{-\frac{1}{2}\alpha(\frac{\partial y_{2,k}}{\partial x_{2,k-1}}+1) - \frac{1}{2}\beta_1\frac{\partial y_{1,k}}{\partial y_{2,k-1}} + \eta[(\frac{3}{2}y_{1km}^2+\mu x_{1km}y_{1km}+\frac{1}{2}x_{1km}^2)\frac{\partial y_{1,k}}{\partial y_{2,k-1}}\right.$$
$$\left. +(\frac{1}{2}\mu y_{1km}^2+x_{1km}y_{1km}+\frac{3}{2}\mu x_{1km}^2)\frac{\partial x_{1,k}}{\partial y_{2,k-1}}] - \frac{1}{2}\gamma\frac{\partial x_{1,k}}{\partial y_{2,k-1}}\right\}.$$

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
