# Peer review of "Nonlinear Vibration Characteristics and Bifurcations of a Rotor System Subjected to Brush Seal Forces"

_applsci, doi:10.3390/app132011539_

Round 1
Reviewer 1 Report
The main problem of this paper is the lack of mathematical rigor. How do the authors derive system (1)? What can they say about well-posedness of the problem?
Moreover, with respect to the stability analysis, details are completely missing. But also some straightforward computations that may help the reader. Is there at least one proof?
Finally, which is the main novelty of their research? And future research perspectives?
I suggest to extensively revise the manuscript, since I do not think it deserves publication in its current form.
The english language needs revisions. Moreover, some typos are present throughout the pages.
Author Response
For the detialed revise report, please see the attached file.

Reviewer 2 Report
I recommend first presenting the numerical simulations (section 4), and then the bifurcation analyses, for a better compression of the dynamics shown therein (section 3). Could the authors discuss the resonant and natural frequency of the system? For minor coments see the file.

Minor editing of English language required
Author Response

(The authors gave the same response as above.)

Reviewer 3 Report
Scientific report on the manuscript: applsci-2639771
Title: Nonlinear Vibration Characteristics and Bifurcations of a Rotor System Subjected to Brush Seal Forces
In this work, nonlinear vibration characteristics of a rotor system are investigated. Such nonlinear rotor system is subjected to brush seal forces which are obtained by integrating the single bristle force along the entire ring.
However, the following suggestions must be taken into account.
1. The language should be reviewed very carefully to correct grammatical errors.
2. Authors should clarify this manuscript's novelty statement and briefly appear the advantage in the Abstract and Conclusion sections.
3. The language should be reviewed very carefully to correct grammatical errors.
4. The introduction should be divided into several subsections; such as Background, Formulation of the Problem of Interest for this Investigation, Literature Survey, Scope and Contribution of this Study, Organization of the Paper.
5. Authors should improve the introduction by including the recent works which deals with same procedure for different models such as
· https://doi.org/10.1007/s42417-022-00808-1
6. Have the authors employed any assumptions on their investigation? Please explain briefly.
7. The components of the Jacobian matrix in Eq. (12) must be included in an appendix.
8. In line 140, the authors write “And equation (13) can be simplified as”. This required that Eq. (13) be before this sentence, while No. (13) comes after this sentence.
9. The obtained results must be compared with the results of previous studies.
10. The concluding remarks must be rearranged to include the important results.
Simple mathematical manipulations are under responsibility of the authors.
The language should be reviewed very carefully to correct grammatical errors.
Author Response

(The authors gave the same response as above.)

Round 2
Reviewer 1 Report
The Authors have extensively revised the paper according to suggestions. I think it now may deserve publication in the Journal.
Reviewer 2 Report
The authors have responded to my comments satisfactorily.
Reviewer 3 Report
The Authors have addressed all my concerns, and I think that the paper can be accepted for publication.